# Molecular Docking, DFT Calculations, Effect of High Energetic Ionizing Radiation, and Biological Evaluation of Some Novel Metal (II) Heteroleptic Complexes Bearing the Thiosemicarbazone Ligand

**DOI:** 10.3390/molecules26195851

**Published:** 2021-09-27

**Authors:** Ehab M. Abdalla, Safaa S. Hassan, Hussein H. Elganzory, Samar A. Aly, Heba Alshater

**Affiliations:** 1Chemistry Department, Faculty of Science, New Valley University, Alkharga 72511, Egypt; ehababdalla99@sci.nvu.edu.eg; 2Chemistry Department, Faculty of Science, Cairo University, Giza 12613, Egypt; hsafaa@sci.cu.edu.eg; 3Department of Chemistry, College of Science, Qassim University, Buraidah 51452, Saudi Arabia; 4Department of Environmental Biotechnology, Genetic Engineering and Biotechnology Research Institute, University of Sadat City, Sadat City 32958, Egypt; samar.mostafa@gebri.usc.edu.eg; 5Department of Forensic Medicine and Clinical Toxicology, University Hospital, Menoufia University, Shebin El-Kom 32511, Egypt; heba.alshater@med.menofia.edu.eg

**Keywords:** DFT calculations, γ–irradiation, anticancer, complexes, docking, thiosemicarbazone

## Abstract

New Pb(II), Mn(II), Hg(II), and Zn(II) complexes, derived from 4-(4-chlorophenyl)-1-(2-(phenylamino)acetyl)thiosemicarbazone, were synthesized. The compounds with general formulas, [Pb(H_2_L)_2_(OAc)_2_]ETOH.H_2_O, [Mn(H_2_L)(HL)]Cl, [Hg_2_(H_2_L)(OH)SO_4_], and [Zn(H_2_L)(HL)]Cl, were characterized by physicochemical and theoretical studies. X-ray diffraction studies showed a decrease in the crystalline size of compounds that were exposed to gamma irradiation (γ-irradiation). Thermal studies of the synthesized complexes showed thermal stability of the Mn(II) and Pb(II) complexes after γ-irradiation compared to those before γ–irradiation, while no changes in the Zn(II) and Hg(II) complexes were observed. The optimized geometric structures of the ligand and metal complexes are discussed regarding density functional theory calculations (DFT). The antimicrobial activities of the ligand and metal complexes against several bacterial and fungal stains were screened before and after irradiation. The Hg(II) complex has shown excellent antibacterial activity before and after γ-irradiation. In vitro cytotoxicity screening of the ligand and the Mn(II) and Zn(II) complexes before and after γ-irradiation disclosed that both the ligand and Mn(II) complex exhibited higher activity against human liver (Hep-G2) than Zn(II). Molecular docking was performed on the active site of MK-2 and showed good results.

## 1. Introduction

Thiosemicarbazones (TSCs) are a class of inorganic metal chelators that exhibit various complexes with transition metals, including Cu, Pd, Ni, and others [1,2]. TSCs and their metal complexes have interesting chemistry because of their variable bonding modes, promising biological implications, structural diversity, and ion-sensing ability [3]. In addition, they have been used as drugs and possess a wide range of biological activities, such as antibacterial [4], antifungal [5,6], antiviral [7], antiamoebic [8], antimalarial [9], and antitumor [10] effectiveness. Numerous literature studies have presented that the biological properties of metal complexes with NS, ONS, and ONN chelating thiosemicarbazones derivatives using various carbonyl compounds, and its complexes [11], are analogs to the metallo-salen compounds of the O,N,N,O-chelating set and have biologically active structures. Antimicrobial activity of vanillin-4-methyl-4-phenyl-3-thiosemicarbazone complexes with cobalt(II), nickel(II), copper(II), and zinc(II) metal ions were examined against Gram-positive, Gram-negative bacteria, and two fungal pathogens. Copper and zinc complexes of vanillin thiosemicarbazone have higher antibacterial and antifungal activities than other complexes [12]. Previous studies manifested the impact of γ-irradiation on the thermal decomposition, spectral, X-ray diffraction, and surface morphology of complexes of transition metals [13]. Furthermore, γ-radiation can influence the color, chemical composition, and catalytic properties, as well as the magnetic, structural, optical, electrical, thermal, and biological activities of sorts of solids [14]. The crystallinity, surface area, particle size, and position and intensity of the characteristic bands in FT-IR and electronic spectral studies of solid materials were shown to be altered [15,16]. Several studies showed the alteration in the antibacterial activity of different tested materials upon exposure to γ-irradiation [17]. Here, we report the synthesis and characterization of complexes of Pb(II), Mn(II), Hg(II), and Zn(II) with the ligand 4-(4-chlorophenyl)-1-(2-(phenylamino) acetyl)thiosemicarbazone. The aim of this work is to study the effect of γ-irradiation on the ligand and its complexes. The FT-IR, UV-Vis absorption spectra, TG/DTG, and X-ray diffraction patterns were studied. The antibacterial, antifungal, and anti-cancer activities of the ligand and its metal complexes before and after γ-irradiation were evaluated.

## 2. Experimental

### 2.1. Material and Methods

All organic compounds and solvents were purchased from Fluka or Merck, Naser City, Egypt. The metal salts Pb(CH_3_COO)_2_, MnCL_2_, HgSO_4_, and ZnCL_2_ were obtained from Fluka and then used for complex synthesis without further purification.

### 2.2. Synthesis of Metal Complexes

Different metal complexes were synthesized by stoichiometric addition of an ethanolic solution of metal salts (MX_2_) (where M = Pb(II), Mn(II), Hg(II), or Zn(II), and X = Cl, SO_4_, or CH_3_COO) and ethanolic solution of 4-(4-chlorophenyl)-1-(2-(phenylamino)acetyl)thiosemicarbazone, which have been previously reported [18]. The mixture was magnetically stirred at 60 °C for 6–8 h. The obtained precipitates were filtered, washed with anhydrous diethyl ether, and then dried under a vacuum in the presence of P_4_O_10_ to afford the complexes (B_1_–B_4_), as shown in Figure 1.

### 2.3. Physical Measurement

Elemental analyses (C, H, and N) were carried out at the Microanalytical Unit, Cairo University, Egypt. Metal content complexometric titration was estimated using EDTA following the standard literature methods, as reported by Basset et al. [19]. The FT-IR spectra were noted as KBr pellets using a Nenexeus-Nicolidite-640-MSAFT-IR spectrometer (4000–400 cm^−1^). Mass spectra were acquired using the electron impact (EI) ionization technique at 70 eV using a Hewlett–Packard MS-5988 GC–MS instrument at the Microanalytical Center, National Research Centre, Dokki, Cairo, Egypt. The UV–visible absorption spectra were measured in an N, N-dimethylformamide (DMF) solution (10^−3^ M) using a 4802 UV/vis double beam spectrophotometer (Dayton, NJ, USA). The molar conductivity measurements were recorded using a Tacussel conductometer type CD6N in DMF solution (10^−3^ M). The magnetic properties of all complexes were recorded at room temperature by the modified Gouy method using a Magnetic Susceptibility Johnson Matthey Balance. The effective magnetic moments were calculated using the relation µeff = 2.828 (X_m_T)^1/2^ B.M., where X_m_ is the molar magnetic susceptibility corrected for diamagnetism of all atoms in the compounds using Selwood and Pascal’s constants. Thermal analysis (TGA/DTG) was obtained by using a Shimadzu DTG-50 Thermal analyzer with a heating rate of 10 °C/min in a nitrogen atmosphere with the rate of 20 mL/min in a temperature range of 25–800 °C using platinum crucibles at the Central Lab, Faculty of Science, Menoufia University, Egypt. Using the Rigku Model ROTAFLEX Ru-200, X-ray diffractograms of the solid samples were measured at the National Research Centre, Cairo, Egypt. Structural analysis of the X-ray diffractograms given by computer control formally was finished using a Philips X’Pert MPD X-ray diffractometer ready with Cu radiation Cu Ka (k = 1.540 56 Å). Usually, the most powder diffractometers use the Bragg–Brentano parafocusing geometry. The X-ray tube was utilized for a copper tube operating at 40 kV and 30 mA. The scanning range (2θ) was 20–80° with a step size of 0.02° and a counting time of 3 s/step. Quartz was used as the standard material to calibrate the instrumental extension. This identification of the complexes was done using the method described by Nair and Appukuttan [20] from the fit identified by the Scherrer formula; the average crystallite size, L, is measured as L = ƛK = ßcosθ, where ƛ is the X-ray wavelength in the manometer, K is a constant equal to 0.9 related to crystallite shape, and ß is the peak width at half maximum height. The value of ß in the 2θ axis of the diffraction shape must be in radians. The θ is the Bragg angle in radians since Cos θ is compatible with the same number.

### 2.4. *Computational Study*

In this part, we tried to discover the optimized geometrical parameters, such as bond lengths, bond angles, and net charges, on the coordinated atoms. The total energies (highest occupied molecular orbital (HOMO) energies, lowest unoccupied molecular orbital (LUMO) energies, and the dipole moments) for the ligand and complexes were computed. Density Functional Theory (DFT) at levels B3LYP, 6-311G, and LANL2DZ as the basis sets are used in all ligands and complexes calculations, respectively [21]. These calculations were carried out using G09W software [22]. All docking steps were done by Molecular Operating Environment (MOE 2008) software (Chemical Computing Group Inc., Montreal, QC, Canada) [23] to simulate the binding model of these compounds into an MK-2 enzyme (3WI6).

### 2.5. Irradiation Studies

For the irradiation studies, the solid samples of prepared compounds (B, B_1_, B_2_, B_3_, and B_4_) were subjected to a 60 kGy γ-irradiation dose with a rate equal to 2.2 kGy h^−1^ [24]. The test was performed using the Indian ^60^Co γ-ray cell type GE-4000 A at room temperature (at the Egyptian Atomic Energy Authority (EAEA), Nasr City, Egypt). After removing the samples from the radiation field, the FT-IR, absorption spectra, thermal analysis (TG/DTG), anti-bacterial, and anti-cancer activities of the irradiated samples were investigated. X-ray powder diffraction analyses of the un-irradiated (B, B_2_, and B_4_) and irradiated (A_2_ and A_4_) samples of the synthesized compounds were conducted using a Rigku Model ROTAFLEX Ru-200 at the National Research Centre, Dokki, Cairo, Egypt. The divergence and receiving slits values were 1 and 0.1, respectively.

### 2.6. Antimicrobial Assay

The antimicrobial activity of the synthesized compounds was determined using the disc-agar diffusion method [25] at the microanalytical unit of Cairo University, Egypt. The antimicrobial activity was performed against the sensitive organisms, including *Escherichia coli* (Gram-negative bacteria), *Staphylococcus aureus* (Gram-positive bacteria), *Aspergillus flavus* (fungi), and *Candida albicans* (fungi). Standard discs of ampicillin (antibacterial agent) and amphotericin B (antifungal agent) were served as positive controls for antimicrobial activity, while the filter discs impregnated with 10 μL of the solvent dimethyl sulfoxide (DMSO) were used as a negative control. Briefly, 100 μL of the test bacteria/fungi was grown in 10 mL of fresh media until they reached a count of approximately 10^8^ cells/mL for bacteria or 10^5^ cells/mL for fungi [26]. A total of 100 μL of microbial suspension was spread onto agar plates corresponding to the broth, in which they were maintained. The plates were inoculated with filamentous fungi (*Aspergillus flavus*) at 25 °C for 48 h; Gram (+) bacteria (*Staphylococcus aureus*) and *Gram* (-) bacteria (*Escherichia coli*) were incubated at 35–37 °C for 24–48 h, and yeast (*Candida albicans*) incubated at 30 °C for 24–48 h. Subsequently, the diameters of the inhibition zones (in millimeters) were measured. Blank paper disks (Schleicher & Schuell, Spain, Sigma-Aldrich, St. Louis, MO, USA) with a diameter of 8.0 mm were impregnated with 10 μL of the tested concentration of the stock solutions. When a filter paper disc, impregnated with a tested chemical, is placed on agar, the chemical will diffuse from the disc into the agar. This diffusion will locate the chemical in the agar only around the disc. Inhibition of the organisms, which is evidenced by the clear zone surrounding each disk, was measured and used to calculate the mean of the inhibition zone.

### 2.7. Cytotoxicity Assays

The cytotoxicity sample was prepared on cells by inoculating a 96-well tissue culture plate with 1 × 10^5^ cells/mL (100 µL/well), incubated at 37 °C for 24 h, with washing media; the cell monolayer was washed twice to improve the complete monolayer sheet, and the growth medium was poured from 96-well microtiter plates after confluent sheet of cells were formed. Two-fold dilutions of the tested sample were prepared in Roswell Park Memorial Institute (RPMI) medium with 2% serum (maintenance medium). Then, 0.1 mL of each dilution was tested in different wells, leaving 3 wells as the control, receiving only a maintenance medium, and the plate was incubated at 37 °C and subsequently examined. Cells were checked for any actual indications of harmfulness, e.g., incomplete or complete loss of the monolayer, rounding, shrinkage, or cell granulation. The 3-(4,5-dimethylthiazol-2-yl)-2,5-diphenyl tetrazolium bromide (MTT) solution was prepared (5 mg/mL in PBS) (Bio Basic Inc., Markham, ON, Canada), and then 20 µL MTT solution was added to each well. To thoroughly mix the MTT into the media, samples were incubated at 37 °C, 5% CO_2_, for 1–5 h to allow the MTT to be metabolized, and then placed on a shaking table at 150 rpm for 5 min, dumping off the media (to remove the dry residue on the paper towels if necessary). Formazan (MTT metabolic product) was resuspended in 200 µL DMSO solution and placed on a shaking table at 150 rpm for 5 min to mix the formazan into the solvent thoroughly. Finally, the optical density was recorded at 560 nm and the subtract background was measured at 620 nm. The optical density should be directly correlated with cell quantity [27,28].

## 3. Results and Discussion

### 3.1. Physicochemical Properties

The experimental results showed that all the synthesized metal complexes are colored, stable in air, and insoluble in most organic solvents, except DMF and DMSO. The elemental analysis and physical data are summarized in Table 1. Elemental analyses indicate that the complexes (B_1_, B_2_ and B_4_) are formed in a 1:2 (M:L) molar ratio while B_3_ is formed in 2:1 (M:L). The analytical results agree well with the suggested formula. The molar conductivity in the 10^−3^ M DMF solution indicated the non-electrolyte nature of the B_1_ and B_3_ complexes while showing the electrolyte nature of the B_2_ and B_4_ complexes [29].

### 3.2. FT-IR

The FT-IR spectra of ligand H_2_L(B, A) and Pb(II), Mn(II), Hg(II) and Zn(II) complexes before and after γ-irradiation (B_1_–B_4_ and A_1_–A_4_) are reported in Table 2, Figure 1, and Appendix A. The FT-IR spectrum of the ligand before irradiation shows stretching frequencies of ν(N4), ν(N2), ν(N1), ν(C=O), and ν(C=S) at 3335, 3302, 3100, 1670, and 750 cm^−1^, respectively. After γ-irradiation, the corresponding bands ν(N4), ν(N2), ν(N1), and ν(C=S) were shifted to higher wave numbers as compared to the free ligand. At the same time, the functional group of ν(C=O) shifted to a lower wave number after γ-irradiation [30].

#### 3.2.1. IR Spectra of the Pb(II) Complexes

The IR spectra of the Pb(II) complexes before and after γ-irradiation showed strong bands at 3437; 3442, 3242; 3296, 3101; 3100, 1667; 1672, and 756; 755 cm^−1^ before and after γ-irradiation, which documented the stretching frequencies of the υ(N4), υ(N2), υ(N1), υ(C=O), and υ(C=S) wagging vibrations, respectively. The IR spectra of the Pb(II) complexes before and after γ-irradiation were clear in that the band corresponding to υ(N1) and υ(C=S) are shifted to a lower frequency and the band corresponding to υ(N4), υ(N2), and υ(C=O) are shifted to a higher frequency after γ-irradiation, with increasing sharper bands after γ-irradiation. The new bands appeared at 625; 548 and 508; 468 cm^−1^, assigned to υ(Pb-O) and υ(Pb-N), respectively. Further, the Pb(II) complexes showed the additional stretching vibration band due to acetate at 1552 and 1595 cm^−1^ before and after γ-irradiation, suggesting uncoordinated acetate ions.

Infrared spectra indicated that the intensity of the bands is sharper by using gamma rays [24].

#### 3.2.2. IR Spectra of Mn(II) Complexes

The IR spectra of the Mn(II) complexes before and after γ-irradiation showed strong bands at 3461; 3459, 3295, 3134; 3100, 1672, and 754; 755 cm^−1^, which are attributed to the stretching frequencies of the υ(N4), υ(N2), υ(N1), υ(C=O), and υ(C=S) wagging vibrations, respectively. It is clear that the band after γ-irradiation corresponding to υ(N4) and υ(N1) is shifted to a lower frequency, and the band corresponding to υ(C=S) is shifted to a higher frequency, which is sharper than the bands after γ-irradiation. The new bands appeared at 637; 508 and 507 cm^−1^, assigned to υ(Mn-O) and υ(Mn -N), respectively.

#### 3.2.3. IR Spectra of Hg (II) Complexes

The IR spectra of the Hg(II) complexes before and after irradiation showed strong bands at 3400, 3295; 3291, 3104, 1671; 1670, and 754 cm^−1^, which are ascribed to the stretching frequencies of the υ(N4), υ(N2), υ(N1), υ(C=O), and υ(C=S) wagging vibrations, respectively. We observed that the band after γ-irradiation corresponding to υ(N4) is shifted to a higher frequency, while the band corresponding to υ(N2), υ(N1), and υ(C=O) is shifted to a lower frequency, with increasing sharper bands after γ-irradiation. The new bands appeared at 602; 606 and 509; 510 cm^−1^, assigned to υ(Hg-O) and υ(Hg-N), respectively, in addition to the SO_4_ group that appears at 1055 and 1054 cm^−1^ before and after irradiation, respectively.

#### 3.2.4. IR Spectra of Zn(II) Complexes

The IR spectra of the Zn(II) complexes before and after irradiation showed strong bands at 3450; 3447, 3295, 3102; 3101, 1671 and 754, 755 cm^−1^, which are attributed to the stretching frequencies of the υ(N4), υ(N2), υ(N1), υ(C=O), and υ(C=S) wagging vibrations, respectively. We observed the band after γ-irradiation corresponding to υ(N4) and υ(N1) is shifted to a lower frequency and the band corresponding to υ(C=S) is shifted to a higher frequency, with sharper bands after γ-irradiation. The new bands appeared at 637; 547 and 550 cm^−1^, assigned to υ(Zn-O) and υ(Zn-N), respectively.

### 3.3. Electronic Spectral Bands

The electronic spectral bands of the ligand and Pb(II), Mn(II), Hg(II), and Zn(II) complexes before (B, B_1_, B_2_, B_3_, and B_4_) and after (A, A_1_, A_2_, A_3_, and A_4_) γ-irradiation (λ_max_, nm) in DMF solution in the range of 200–800 nm at room temperature are listed in Table 3 and Figure 2. UV spectra of the ligand (B and A) were perceived as the existence of two absorption bands at 260, 300, and 266, 303 nm before and after γ-irradiation, respectively, assigned to the π–π* transition. While, the UV-visible spectra of the Mn(II) complexes (B_2_ and A_2_) before and after irradiation exhibited bands at 294, 292, 378, and 376 nm, respectively. The Zn(II) complexes (B_4_ and A_4_) before and after irradiation also displayed bands at 291, 294, 375, and 378 nm, which may be assigned to n-π* transitions, representing a square planar geometry [31,32]. The electronic spectra of the Hg(II) complexes before and after irradiation (B_3_ and A_3_) resulted in bands at 276, 281, 372, and 376 nm [33,34]. On the other side, the electronic spectra of the Pb(II) complexes (B_1_ and A_1_) show three-bands at 270, 300, 320, and 354 nm assigned to n-π* transitions in an octahedral geometry around the Pb(II) ion, which is further confirmed by its diamagnetic nature [35,36]. The complexes showed no d-d band; the complexes contained only paired electrons and were diamagnetic.

After **γ**-irradiation, all peaks presented in the spectral diagram were observed. The difference between the electronic spectra of the ligand and its complexes in changing the value of the λ_max_ position and the value of absorbance were also investigated, indicating no change in the complexes’ geometry; this result agrees with previous related studies. Irradiation can induce perturbation of energy levels as well as a deformity in the molecule [17].

### 3.4. PXRD of the Ligand and Metal Complexes

The X-ray diffractograms of the ligand (B) and the Mn(II) (B_2_, A_2_) and Zn(II) (B_4_, A_4_) complexes were evaluated (Table 4 and Figure 3 and Appendix A). The powder diffraction patterns were recorded over the 2θ = 5–90 range lattice constants. The intensities of the powder lines and the corresponding 2θ values are different for the irradiated samples, indicating the amorphous nature of the complex, whereas, upon irradiation. the sample changed to crystalline materials. The average particle size of the crystalline structure of the ligand (B) and its complexes before and after irradiation was calculated using Scherer’s equation [27,36]. The Scherer’s constant (K) in the formula refers to the particle’s shape, and it generally has a value of 0.9. It was found that the calculated crystalline size was in the nano range. The crystallite sizes were 35.12, 14.15, 32.59, 46.7, and 52.7 nm for the ligand, Mn(II), and Zn(II) unirradiated (B, B_2_, B_4_) and irradiated (A_2_, A_4_), respectively. The change in the size of the crystals may be due to the stress induced by the irradiation.

### 3.5. Mass Spectra

The mass spectra were mainly performed to confirm the composition and support the structure of the synthesized chelates. The molecular ion peaks for the Hg(II) complex were examined at *m/z* = 849.1 (Table 1 and Figure 4), which suggests the stoichiometry of the metal and ligand in metal chelates as a 2:1 ratio. The observed data of the complexes are in agreement with their formula as designated from the microanalytical data.

The proposed structures of the complexes were established from analytical and multi spectroscopic methods (Figure 1).

### 3.6. Thermal Analysis

The thermal behavior of the ligand (B, A) and Pb(II) (B_1_, A_1_), Mn(II) (B_2_, A_2_), Hg(II) (B_3_, A_3_), and Zn(II) (B_4_, A_4_) complexes before and after γ-irradiation are listed in Table 5 and Figure 5a–e. The TG curves of B and A in Figure 5a revealed thermal stability till 190 °C and 240 °C, and also showed two decomposition steps in a temperature range of 190–633 °C and 240–680 °C, with the total weight loss of calc. 99.9% (found 100%) and calc. 100% (found 100%) before and after γ-irradiation, respectively. After irradiation, the thermogravimetric analyses (TG) curves of the ligand revealed that γ-irradiation stimulated more thermal stability of the substance than those that were un-irradiated. These results are consistent with the structure of the ligand resolute as determined by the elemental analysis and IR spectroscopy.

Figure 5b shows that the TG curves of the Pb(II) complexes before B_4_ and after A_4_ γ-irradiation exhibited weight loss calc. 6.047% (found 6.75%) at a temperature range of 30–125 °C, attributed to losing one molecule of water and ethanol on one step. On further heating, the complexes decomposed at 245–548 °C. The final stage ended with the remaining Pb as the final residue. After γ-irradiation, the TG curve of (A_4_) was similar to (B_4_) before γ-irradiation.

Figure 5c shows TG curves for the Mn(II) complexes before B_1_ and after A_1_ γ-irradiation. The TG curves of B_1_ exhibited three degradation stages; the first stage in a temperature range of 170–402 °C exhibited weight loss (calc./found % 62.34/62.31) allocated to the loss of the C_24_H_22_Cl_3_N_4_ moiety. The second stage in a temperature range of 436–553 °C (calc./found % 13.19/13.22) was correlated with the loss of the C_3_H_4_N_2_S moiety. The third stage in a temperature range of 719–791 °C (calc./found % 11.35/11.32) corresponded to the loss of the C_2_H_2_N_2_S moiety, leaving species MnO + CO in the fourth stage as the final remainder over at 791 °C. Moreover, the TG curve of A_1_ displayed three degradation stages; the first stage in a temperature range of 172–329 °C exhibited weight loss (calc./found % 77.64/77.62) that was allocated to the loss of the C_28_H_26_Cl_3_N_4_S_2_ moiety. The second stage in a temperature range of 350–652 °C (calc./found % 9.23/9.21) corresponded to the loss of the CH_2_N_4_ moiety, leaving species of MnO + CO in the third stage as the final remainder over at 652 °C.

Figure 5d shows TG curves for the Hg(II) complexes before B_3_ and after A_3_ γ-irradiation. The TG curve of B_3_ exhibited five degradation stages; the first exhibited weight loss (calc./found % 32.68/32.65) in a temperature range of 117–194 °C, which was allocated to the loss of the C_14_H_14_ClN_2_S moiety. The second stage in a temperature range of 194–287 °C (Calc./Found % 1.64/1.60) corresponded to the loss of CH_2_. In addition, the third stage in a temperature range 296–421 °C (calc./found % 11.30/11.32) was correlated with the loss of SO_4_. The fourth stage in a temperature range of 429–615 °C (calc./found % 3.53/3.55) corresponded to the loss of NO, leaving metal oxide (2HgO) as the final residue over at 615 °C. The TG curve after irradiation A_3_ was similar to before irradiation B_3_,with a different extent of dissociation.

Figure 5e shows TG curves for the Zn(II) complexes before B_2_ and after A_2_ γ-irradiation. It observed that the TG curves of B_2_ were similar to that of A_2_, and both of them exhibited four degradation stages; the first stage exhibited weight loss (calc./found % 55.76/55.74) in a temperature range of 152–356 °C, which was allocated to the loss of the C_23_H_18_N_2_Cl_3_ moiety. The second stage in a temperature range of 482–652 °C (calc./found % 22.94/22.92) corresponds to the loss of the C_4_H_8_N_4_S_2_ moiety. While, the third stage in a temperature range of 733–799 °C (calc./found % 5.47/5.50) was correlated with the loss of the CH_2_N_2_ moiety, leaving Zn + 2CO in the fourth stage as the final remainder over at 799 °C.

### 3.7. DFT Calculations of the Ligand and Metal Complexes

The natural charges, obtained from Natural Bond Orbital Analysis (NBO), showed that the more negative active sites were in the following order: S14 (−0.29189) < N12 (−0.40986) < O10 (−0.65733). Thus, the metal ions preferred the coordination through O10, N12, or S14, forming membered rings.

Appendix A shows the optimized structures of the ligand and metal complexes as the lowest energy configurations. The lead atom is six-coordinate in an octahedral geometry by using the S14, N12, and O-acetate donor atoms. However, the other metal complexes were optimized in tetrahedral shapes using the most electron-donating atoms S14 and N12. Many bond lengths were elongated after coordination that supported the coordination through the previously mentioned donor sites. Some of these bond lengths were elongated, such as R(C13-N15), R(N12-C13), R(N12-N11), R(C9-O10), and R(C8-N7) [37]. The polarity of ligand increased after complexation by its coordination with metal ions, as indicated from the magnitude of their dipole moments. The ionic complexes have higher polarity than the non-electrolytic complexes. The natural charges computed from the NBO analysis on the coordinated atoms are observed in Table 6 as Hg (+0.852), Mn(+0.650), Pb(+1.678), and Zn(+1.031). For donor centers, the examined ligand was changed to less negative values, as [S14(−0.048), N12(−0.217), O10(−0.404), and O24(−0.830)] in the Hg chelate; [N12(−0.320) and O10(−0.404)] in the Mn chelate; [N12(−0.367), O10(−0.497), and O47(−0.616)] in the Pb chelate; and [N12(−0.335) and O10(−0.461)] in the Zn chelate. Thus, it can be explained due to the charge transfer from the examined ligand donor sites to the central metal ions, i.e.,
L→M. Table 7 shows some of the optimized bond lengths (Å) and bond angles (degrees) for H_2_L and the complexes using B3LYP/6-311G. The coordinates of the optimized ligand and complexes are tabulated in Appendix A.

The theoretically calculated frontier molecular orbitals in the ground state are illustrated in Figure 6. The global reactivity descriptors, such as the ionization potential (I), electron affinity (A), absolute electronegativity (χ), absolute hardness (η), and softness (S), for the molecule were calculated at the same levels, and the results are presented in Table 8. These molecular properties can be calculated as follows: hardness, η = (I − A)/2; softness (S), S = 1/2η; chemical potential (μ), μ = −(I + A)/2; and electronegativity (χ), χ = (I + A)/2

The lower value of the energy gap explains the charge transfer interactions taking place within the molecule, which influences the molecule’s biological activity. The energy gap reflects the chemical activity of the molecule. A molecule with a small frontier orbital gap is generally associated with a high chemical reactivity and is defined as a soft molecule.

### 3.8. Biological Applications

#### 3.8.1. Antimicrobial Activity

The thiosemicarbazide ligand and chelate compounds before and after irradiation had varying degrees of inhibitory effect on the growth of Gram-negative bacteria, such as *Escherichia coli*, *Klebsiella pneumonia*, and *Pseudomonas aeruginosa*, Gram-positive bacteria (*Staphylococcus aureus* and *Streptococcus mutans*), and fungi (*Candida albicans* and *Asperagillus Nigar*); the inhibitory effect results are presented in Appendix A and Figure 7. Furthermore, *C. albicans* was affected by complex (B_3_, B_4_) before irradiation, with inhibition ranges of 10.3 ± 0.5 and 23.6 ± 0.6, respectively. In turn, complex (A_3_) after irradiation showed higher activity, with an inhibition range of 31.6 ± 0.6. *A. nigar* was affected by complex (B_3_) before irradiation, with an inhibition range of 29.3 ± 0.6, and the higher activity was for complex (A_3_) after irradiation, with an inhibition range of 30.6 ± 0.6; this is compared with the positive control drug used for both fungi. The in vitro antimicrobial activity exhibited by the synthesized compounds before and after irradiation is in Appendix A. The results showed that complex (B_3_) before irradiation towards Gram-negative bacteria (*E. coli, K. pneumonia,* and *P. aeruginosa*) had higher activity than other complexes, with an inhibition range of 23.3 ± 0.6, 22.6 ± 0.6, and 21.3 ± 0.6, respectively. The complex (A_3_) after irradiation had the highest activity, with an inhibition range of 29.6 ± 0.6, 20.6 ± 0.6, and 27.6 ± 0.6, respectively, compared with other complexes before and after irradiation. In addition, the results reported that complex (A_3_) after irradiation had the highest activity with an inhibition range of 33.3 ± 0.6 and 19.6 ± 0.6, while complex (B_3_) before irradiation towards Gram-positive bacteria (*S. aureus* and *S. mutans*) had higher activity than other complexes, with an inhibition range of 36.6 ± 0.6 and 28.6 ± 0.6, respectively. Therefore, the [Hg_2_(H_2_L)(OH)SO_4_] complex after irradiation had a higher activity than other complexes. These results can be demonstrated according to the basis of Overtone’s concept and Tweedy’s chelation theory [38,39,40,41], as the chelation increases the delocalization of p-electrons over the whole ring. Hence, this enhances the compounds’ penetration into lipid membranes. In addition, the oxidation state of the metal ion, type, and number of donor sites besides their relative presence within the ligand, solubility, conductivity, particle size, and bond length between the metal and ligand are also important in determining the antimicrobial activity of compounds [42,43].

#### 3.8.2. Cytotoxicity

The cytotoxic activities of the ligand and their complexes before and after irradiation were evaluated against the human liver cancer cell line (HepG2) and normal cell line (HEK-293), as presented in Table 9, Figure 8, and Appendix A. The results are expressed as the IC_50_, which is the concentration of a drug that causes a 50% reduction in the proliferation of cancer cells when compared to the growth of the control cells. The thiosemicarbazide ligand before irradiation (B) was more biologically active than after irradiation (A), where the IC_50_ value of B is 20.45, while A is 29.25. The Mn(II) and Zn(II) complexes after irradiation (A_2_, A_4_) against the human liver HepG2 cancer are more effective than the Mn(II) and Zn(II) complexes before irradiation (B_2_, B_4_). Moreover, (A_2_, A_4_) had lower IC_50_ values than (B_2_, B_4_), respectively.

The attained IC_50_ values of vinblastine, the ligand, and Mn(II) and Zn(II) complexes before and after irradiation are in the following order:

Vinblastine (4.58) > B (20.45) > A_2_ (23.95) > A (29.45) > B_2_(32.6) > A_4_ (86.24) > B_4_ (189.96) µg/mL. These obtained results concluded that the synthesized ligand and its complexes have a good anticancer effect on the HepG2 cell line, except the Zn(II) complex before irradiation (B_4_) and Mn(II) complex after irradiation (A_2_), which are the most active ones. Moreover, vinblastine had an enhanced anticancer effect on the selected cancer cell lines.

#### 3.8.3. Molecular Docking Studies

Various thiosemicarbazide derivatives have been used as starting materials for compounds with better biological activities. Molecular modeling tools are used to explore their mechanism of action. One of the most important enzymes that control signal transduction and cell proliferation is mitogen-activated protein kinase-activated protein kinase 2 (MAPKAPK-2 or MK-2) [44]. Discovering new inhibitors of this key enzyme has received attention as a strategy for discovering novel anticancer agents [45]. The ligand and its metal complexes were docked on the active site of the MK-2 enzyme in a trial to suggest a mechanism of action for their cytotoxicity. The protein data bank file is PDB: 3WI6). The file contains an MK-2 enzyme co-crystallized with an inhibitor. All docking procedures were achieved by MOE software. The inhibitor interacts with the MK-2 active site with Glu-145 hydrogen bonds involving Glu-145 and Asp 207 (Figure 9). The docking protocol was validated by redocking the Mn, Zn, and Pb complexes on the active site of the MK-2 enzyme with the highest energy score for the Pb complex (−7.28 kcal/mol). All the docked compounds were fitted on the active site of the MK-2 enzyme. The docking scores and amino acid interactions for the docked compounds are summarized in Table 10. The types of interactions involved were side-chain acceptor, metal contact receptor, and solvent contact. The Hg compound displayed the best docking score (−8.16 kcal/mol), which may explain its promising cytotoxic activity. Finally, we can conclude that the molecular docking of our compounds on the active site of the mitogen-activated kinase (MK-2) revealed good amino acid interactions. We observed good agreement between the experimental IC_50_ values and the molecular docking of the selected enzyme target relative to the Pb and Mn complexes, but the ligand scoring energy sequence was not compatible with its IC_50_ value.

## 4. Conclusions

New Pb(II), Mn(II), Hg(II), and Zn(II) complexes were prepared, and their molecular structures were described before and after γ-irradiation. The results documented the following:The ligand behaves as a monobasic or neutral tetradentate in complexes B_1_, B_2,_ and B_4_, while complex (B_3_) is binuclear.The DFT study showed the suggested geometrical structures of our compounds.Based on the results obtained from the FT-IR spectra of complexes before (B_1_–B_4_) and after (A_1_–A_4_) irradiation, the band of spectra after irradiation was sharper than before.No significant change was detected in the structures, and only a slight shift in the wavelength and the absorbance after the exposure to gamma-ray.According to the powder X-ray results, it was noticed that the calculated crystalline size of the ligand and complexes fell within the nano range.Mn(II) complex after irradiation against the human liver cancer cell line (HepG2) reflected higher anti-cancer activity than the ligand and Zn(II) complexes.The molecular docking showed that all the compounds have a potential antitumor effect, especially the Hg chelate with a more negative scoring energy value, which is expected to inhibit the active site of mitogen activated kinase (MK-2).

## Data Availability

The data presented in this study are available on request from the corresponding author.

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
