# Peer review of "Molecular Docking, DFT Calculations, Effect of High Energetic Ionizing Radiation, and Biological Evaluation of Some Novel Metal (II) Heteroleptic Complexes Bearing the Thiosemicarbazone Ligand"

_molecules, 2021, doi:10.3390/molecules26195851_

Round 1

Reviewer 1 Report

The authors report on molecular docking, DFT calculations, and biological evaluations of metal complexes with thiosemicabazone ligand system. First of all, there are too many typos in the manuscript that it will need to be corrected. Although Thiosemicabazone ligand systems are well known for their reaction to transition metals, it is regrettable that the structures were expected and the DFT calculations were performed without crystal structures. It is believed that securing the X-ray crystal structures and then conducting biological evaluation will lead to a more robust conclusion. The metal complexes proposed in this manuscript are expected to remove cancer cells with toxicity, so toxicity to normal cells should be identified. The results of biological study of normal cells should be added. Additionally, it is better to further explain the results of biological evaluation on Pb(II) and Hg(II) complexes and why B4 compounds exhibit abnormal toxicity.

Author Response

Response:

- Indeed, X-ray crystal structure would be ideal and direct solid evidence for the proposed structure; however, all our efforts to grow single crystals for all prepared complexes were unsuccessful.

- Normal cells added in table 9 and figure S3.

- IC50 of B4 is 89.96 not 189.96 (Misspelled).

- Due to the impact of the COVID-19 pandemic we cannot perform the experiments of biological evaluation on Pb(II) and Hg(II) complexes within the requested timeframe.

Reviewer 2 Report

In this manuscript, the authors synthesized and studied the new Pb(II), Mn(II), Hg(II), and Zn(II) complexes derived from 4-(4-chlorophenyl)-1-(2-(phenylamino)acetyl)thiosemicarbazone. They studied X‐ray diffraction and thermal properties of these compounds, as well as the optimized geometric structures of the ligand and metal complexes by using Natural Bond Orbital Analysis and B3LYP/6-311G(++)d,p density functional theory. They found a decrease of the crystalline size of the compounds exposed to Gamma irradiation, and a thermal stability of the Mn(II) and Pb(II) synthesized complexes after γ-irradiation, compared to those before γ–irradiation, whereas no changes of the Zn(II) and Hg(II) complexes were observed. The authors discussed the optimized geometric structures of the ligand and metal complexes obtained from the DFT calculations, and performed molecular docking on the active site of MK-2 with good results. The authors studied the antimicrobial activities of the ligand and metal complexes against several bacterial and fungal stains before and after irradiation. They found excellent antibacterial activity of the Hg(II) complex, abd that both the ligand and Mn(II) complex exhibited higher activity than the Zn(II) complex against human liver (Hep-G2). 

The studied presented in this manuscript should be of considerable interest to researchers in the area of biomolecular systems. The introduction is adequate, the description of the protocols is sufficient, and the results discussion is satisfactory. The manuscript deserves publication after the authors correct numerous grammar errors in their text.

Author Response

Response:
The manuscript has been carefully checked, the language has been edited by native speakers and all errors have been corrected.

Reviewer 3 Report

The manuscript by Abdalla et al. presents a work on the spectroscopic and computational characterization of complexes between thiosemicarbazone derivatives and Pb(II), Zn(II), Mn(II), Hg(II). Antimicrobial and cytotoxic experiments are also presented. The main interest as presented by the authors is the use of gamma radiation and the evaluation of its effect on the biological activity of the complexes.

The work could be of interest for Molecules but the key element of the work, which is the effects of gamma radiation, is not investigated deeply enough. The presented experimental evidence suggest minor or no changes on the physicochemical properties of the complexes, which do not explain the changes on the biological activities. Also the authors make little effort in explaining this point.

In addition, there are a number of other major issues that should be addressed before consideration for publication.

Finally, the use of the English language must be improved.

Major issues:

English must be improved. Currently, on too many sentences the meaning is not clear or precise. For example, these sentences taken from the conclusions:

  • What is the meaning of conclusion 3 “Irradiated complexes (A1- A4) are more sharper than unirradiated (B1-B4)”? Is this referred to the bands in the spectra? Also revise things like “more sharper”
  • Check the language in “No change in structures, slightly shift of wave length and absorbance by using gamma-ray”. There is no verb in the sentence.
  • Again, the language: “Chelates compound are in Nano range for before and after irradiation” What is the meaning? Is the “Nano range” referred to the cytotoxic activity?

Gamma radiation effects on the complexes

  1. The application of gamma rays on the complexes has a minor effect on the crystallinity and no effect on the chemical composition or structure. However, there are significant changes on the biological activity. In some cases for the better and in some for the worse. What is the reason for the changes in the biological activity? Currently there is a too vague explanation at the end of section 3.8.1 but this point is key for this work. More experimental evidence should be provided to understand the effects of the radiation on each of the used compounds and the authors should make an effort to explain how these translate to the biological activity.

Methods

  1. The authors should indicate how they performed the geometry otpimizations in the DFT calculations. If they performed frequency calculations and vibrational analyses to check if the found structures are energy minima.
  2. In section 3.7 it is said that figures 6ª and 6b-e confirm the optimized structures of the ligand and metal complexes. Maybe this is a deficiency of the used English since by themselves, figures cannot confirm a structure as an energy minimum. The proper method is to perform vibrational analysis on each optimized structure and inspect for the absence of imaginary frequencies.
  3. The coordinates of the optimized ligand and complexes should be included in the supporting information material for reproducibility reasons.
  4. The setup of the NBO analysis should be included in the description of the computational methods.
  5. There should be a description of the method to calculate the molecular descriptors such as electron affinity etc. Which software was used for this purpose?
  6. It is not accurate to say that the DFT calculations supported the suggested geometries (conclusions) because currently the geometries of the complexes are not obtained from the experimental data (e.g. from IR), they are only reported from the DFT calculations.

Basis set:

  1. In the methods section, it is indicated that the 6-311G basis set was used for the ligand. However, in section 3.7, the basis set 6-311++G(d,p) is mentioned. Please make clear which is the employed basis set. Was this basis set used only for the NBO analysis?
  2. Also, it is indicated that the LANL2DZ is used for the complex. Was it used for the entire complex or only for the metal atom?
  3. The correct notation for the basis set is 6-311++G(d,p) and not 6-311G(++)d,p. Please correct.

Docking

  1. It is stated that the compound with larger binding affinity resulting from the docking calculations is the Hg complex (i.e. compound B3). It is also stated that this fact could explain its promising cytotoxic activity. However, there is no data for the cytotoxic activity of B3 on Table 8 or Figure 9. This should be corrected or explained.
  2. In addition, the ligand and compound B2, which show the larger cytotoxic activity (lower IC50), have the lowest binding affinities according to the docking calculations. This suggests that the enzyme target considered for the docking may not be the one on which the compounds execute their activity or that the docking is not performed in the correct binding site or with a different conformation of the enzyme. This should be discussed.
  3. The images showing the protein-complex interactions in figure 10, both in 2D and 3D, are very small do not allow for a clear understanding of the interactions. Please, make the images larger and show clearly the important interactions.
  4. It is not correct to say (see conclusions) that the docking calculations indicate that a compound has good antitumor effect because the docking calculations only provide a rough estimation of the binding affinity of a molecule under many approximations. There are many other factors that make a compound a good or bad drug (i.e. the antitumor effect measured overall).

Minor

  1. Table 2. The IR bands corresponding to NH vibrations are labelled as N1, N2, N4. Can the authors label the atoms N1 , N2 etc in the molecular structures?
  2. Some atoms are labeled (S14, N12, O10) in section 3.7 but the labels in figure 6 are impossible to read. The numbering or labels should be clearly introduced. For example in scheme 1.
  3. The molecular structures in Figure 6 are not clear. This figure should be improved.
  4. Although geometry optimizations provides structures with a very fine precision on the position of the atoms, it is meaningless to report bond distances to the 5th decimal of Angstrom for various reasons such as the numerous approximations made in the quantum theory: time independent Schrodinger equation, DFT approximations, basis set and numerical accuracy of the calculation. 2 decimal places should be enough. Something similar can be applied to the dipole moments reported in table 7.

Author Response

Major issues:

English must be improved. Currently, on too many sentences the meaning is not clear or precise. For example, these sentences taken from the conclusions:

- What is the meaning of conclusion 3 “Irradiated complexes (A1-A4) are more sharper than unirradiated (B1-B4)”? Is this referred to the bands in the spectra? Also revise things like “moresharper”

Response:
replaced by “3-          Based on the results obtained from the FT-IR spectra of complexes before (B1-B4) and after (A1-A4) irradiation, the band of spectra after irradiation was sharper than before”.

- Check the language in “No change in structures, slightly shift of wave length and absorbance by using gamma-ray”. There is no verb in the sentence.

Response: replaced by “No significant change was detected in structures, a slight shift of the wavelength, and the absorbance after the exposure to gamma-ray”.

- Again, the language: “Chelates compound are in Nano range for before and after irradiation” What is the meaning? Is the “Nano range” referred to the cytotoxic activity?

Response: According to powder x-ray results, it was noticed that the calculated crystalline size of ligand and complexes fell within the nano range”.

Gamma radiation effects on the complexes

  1. The application of gamma rays on the complexes has a minor effect on the crystallinity and no effect on the chemical composition or structure. However, there are significant

changes on the biological activity. In some cases for the better and in some for the worse. What is the reason for the changes in the biological activity? Currently there is a too

vague explanation at the end of section 3.8.1 but this point is key for this work. More experimental evidence should be provided to understand the effects of the radiation on each of the used compounds and the authors should make an effort to explain how these translate to the biological activity.

Response:

        We totally agree with the reviewer’s comment in this regard. Despite the γ-irradiation of these compounds have minor change on crystallinity and no effect on their chemical compositions or structures as investigated in this manuscript. Meanwhile, they exhibited significant biological activities. As reported in many previous researches, γ-irradiation exceeds the stability of chemical compounds depending on not only their coordination and structures, but also, the beam intensity of γ-radiation.

To interpret this finding, we have two assumptions:

        - γ- irradiation may increase the stability of the tested compounds, leading to a prolonged duration of these compounds in the biological media causing significant activities.

       - On the other side, γ-irradiation can make the tested compounds more fragile to be dissociated in the biological media and give their activities through some moieties.

Therefore, further studies or more investigations should be conducted to explore the mechanism of action of γ-irradiation.

Methods

  1. The authors should indicate how they performed the geometry otpimizations in the DFT calculations. If they performed frequency calculations and vibrational analyses to check if the found structures are energy minima.

           Response: we calculated frequency and no imaginary values were observed

  1. In section 3.7 it is said that figures 6ª and 6b-e confirm the optimized structures of the ligand and metal complexes. Maybe this is a deficiency of the used English since by themselves, figures cannot confirm a structure as an energy minimum. The proper method is to perform vibrational analysis on each optimized structure and inspect for the absence of imaginary frequencies.

            Response: Figure 6 show the optimized structure of the ligand and metal complexes.

  1. The coordinates of the optimized ligand and complexes should be included in the supporting information material for reproducibility reasons.

Response: added in Table (S1-S5)

  1. The setup of the NBO analysis should be included in the description of the computational methods.

Response: added in Table 6

  1. There should be a description of the method to calculate the molecular descriptors such as electron affinity etc. Which software was used for this purpose?

         Response: The molecular properties are mentioned in (Table 7) which can be   calculated as follows: Hardness η = (I-A)/2, Softness (S) S= 1/ 2η, Chemical potential (μ), μ = -(I +A)/2 and Electronegativity (χ), χ = (I + A)/2

  1. It is not accurate to say that the DFT calculations supported the suggested geometries (conclusions) because currently the geometries of the complexes are not obtained from the experimental data (e.g. from IR), they are only reported from the DFT calculations.

Basis set:

          Response: DFT study showed the suggested geometrical structures of our compounds.

  1. In the methods section, it is indicated that the 6-311G basis set was used for the ligand. However, in section 3.7, the basis set 6-311++G(d,p) is mentioned. Please make clear

which is the employed basis set. Was this basis set used only for the NBO analysis?

         Response: 6-311G basis set was used for ligand with its optimization and NBO analysis

  1. Also, it is indicated that the LANL2DZ is used for the complex. Was it used for the entire complex or only for the metal atom?

            Response: LANL2DZ was used for the entire complex.

  1. The correct notation for the basis set is 6-311++G(d,p) and not 6-311G(++)d,p. Please correct.

           Response: 6-311G basis set only, it is a typing mistake.

Docking

  1. It is stated that the compound with larger binding affinity resulting from the docking calculations is the Hg complex (i.e. compound B3). It is also stated that this fact could explain its promising cytotoxic activity. However, there is no data for the cytotoxic activity of B3 on Table 8 or Figure 9. This should be corrected or explained.

           Response: We expected it’s cytotoxic theoretically only by comparing its scoring energy value with the other compounds.

  1. In addition, the ligand and compound B2, which show the larger cytotoxic activity (lower IC50), have the lowest binding affinities according to the docking calculations. This suggests that the enzyme target considered for the docking may not be the one on which the compounds execute their activity or that the docking is not performed in the correct binding site or with a different conformation of the enzyme. This should be

discussed.

       Response: We observed good agreement between the experimental IC50 values with the molecular docking of the selected enzyme target relative to Pb and Mn complexes but ligand scoring energy sequence not compatible with its IC50 value.

  1. The images showing the protein-complex interactions in figure 10, both in 2D and 3D, are very small do not allow for a clear understanding of the interactions. Please, make the

images larger and show clearly the important interactions.

Response: done

  1. It is not correct to say (see conclusions) that the docking calculations indicate that a compound has good antitumor effect because the docking calculations only provide a rough estimation of the binding affinity of a molecule under many approximations. There are many other factors that make a compound a good or bad drug (i.e. the antitumor effect

measured overall).

Response: The molecular docking showed that all compounds have antitumor potential affect specially Hg chelate with the more its negative scoring energy value that expected to inhibit of the active site of mitogen activated kinase (MK-2)

Minor

  1. Table 2. The IR bands corresponding to NH vibrations are labelled as N1, N2, N4. Can the authors label the atoms N1 , N2 etc in the molecular structures?

Response: done

  1. Some atoms are labeled (S14, N12, O10) in section 3.7 but the labels in figure 6 are impossible to read. The numbering or labels should be clearly introduced. For example in

scheme 1.

          Response: done

  1. The molecular structures in Figure 6 are not clear. This figure should be improved.

Response: done

  1. Although geometry optimizations provides structures with a very fine precision on the position of the atoms, it is meaningless to report bond distances to the 5 decimal of

Angstrom for various reasons such as the numerous approximations made in the quantum theory: time independent Schrodinger equation, DFT approximations, basis set and numerical accuracy of the calculation. 2 decimal places should be enough. Something similar can be applied to the dipole moments reported in table 7.

Round 2

Reviewer 1 Report

This manuscript describes the ionizing radiation and biological evaluation of metal heteroleptic complexes with thiosemicarbazone ligand system. I think it will be help to researchers who study related fields with appropriate molecular docking and DFT calculation and explanations for structural differences caused by changes metals. Overall, I would like to recommend the manuscript for publication.

Minor revisions:

Figure 6 should be moved to supporting information.

Reviewer 3 Report

The authors addressed most of the posed issues, although the explanation of the effects of gamma radiation on the samples is not satisfactory. However, I do not have major reason for not reccommend the work for publication.